# Simplified Attachable EEG Revealed Child Development Dependent Neurofeedback Brain Acute Activities in Comparison with Visual Numerical Discrimination Task and Resting

**DOI:** 10.3390/s22197207

**Published:** 2022-09-23

**Authors:** Kazuyuki Oda, Ricki Colman, Mamiko Koshiba

**Affiliations:** 1Engineering Department, Graduate School of Sciences and Technology for Innovation Yamaguchi University, Yamaguchi 755-8611, Japan; 2Department of Cell and Regenerative Biology, University of Wisconsin, Madison, Madison, WI 53706, USA; 3Department of Pediatrics, Saitama Medical University, Saitama 350-0495, Japan; 4Graduate School of Information Sciences, Tohoku University, Sendai 980-8579, Japan

**Keywords:** easy mounting Bluetooth EEG, Sp1 and Sp2 headset, size control of local mascot, neurofeedback, visual display game, beta/theta ratio, four wave bands

## Abstract

The development of an easy-to-attach electroencephalograph (EEG) would enable its frequent use for the assessment of neurodevelopment and clinical monitoring. In this study, we designed a two-channel EEG headband measurement device that could be used safely and was easily attachable and removable without the need for restraint or electrode paste or gel. Next, we explored the use of this device for neurofeedback applications relevant to education or neurocognitive development. We developed a prototype visual neurofeedback game in which the size of a familiar local mascot changes in the PC display depending on the user’s brain wave activity. We tested this application at a local children’s play event. Children at the event were invited to experience the game and, upon agreement, were provided with an explanation of the game and support in attaching the EEG device. The game began with a consecutive number visual discrimination task which was followed by an open-eye resting condition and then a neurofeedback task. Preliminary linear regression analyses by the least-squares method of the acquired EEG and age data in 30 participants from 5 to 20 years old suggested an age-dependent left brain lateralization of beta waves at the neurofeedback stage (*p* = 0.052) and of alpha waves at the open-eye resting stage (*p* = 0.044) with potential involvement of other wave bands. These results require further validation.

## 1. Introduction

Brain growth is a central issue in child development. Assessment of electroencephalograph (EEG) readings during specific tasks can serve as a method for quantitatively determining the physiological growth state of the brain [1]. Changes in basic EEG rhythms with age have been identified. In an eyes-closed resting condition, age-dependent decreases in delta and theta waves and increases in alpha and beta waves, with decreases in the theta/alpha and theta/beta ratios have previously been reported [2]. The use of EEG to visualize brain activity may prove to be an innovative education and rehabilitation tool during specific developmental periods [3]. Cognitive advancement in children inevitably requires further complex learning. Novel education technologies such as those we have developed that support neurodevelopmental visualization of self-regulating functions in cognition and emotion, such as concentration and relaxation, may play an important role [4].

Neurofeedback technology (NFT) that actively activates brain self-regulating functions by visualizing their own brain activity utilizing EEG [5] has emerged in recent years. NFT has been applied less often in typically developing children, but [6] it has been used clinically in children to both diagnose and provide therapy [7] for ADHD [8], using NFT theta, alpha, sensorimotor (SMR), and beta wave training [9] and for children on the autism spectrum [10] and with learning disorders [11] to improve cognitive function. The beneficial neuronal plasticity effects of NFT education and therapy likely require repetition [12], which means that the NFT system needs to be something easy that children will willingly participate in and can be easily worn at home or school. A simple attachable EEG would enable frequent use for neurodevelopmental support applications in daily education settings and clinical monitoring. Recordings from sphenoidal electrodes have been suggested to improve the localization precision of EEG recordings in patients with temporal lobe epilepsy. However, the usage was limited [13].

To meet these requirements, we designed a two-channel wireless EEG headband device set with electrodes for Sp1 and Sp2 of the 10–20 system [14]. This device can be safely and easily placed on the forehead without the need for restraint or electrode paste and can be easily removed.

We explored the utility of our device for NFT methods that could be applied to education and therapy for neural development by having children wear the device while playing games that should appeal to variously aged children. We developed a visual NFT game prototype in which the local mascot of Hagi city in Yamaguchi prefecture, Japan, called “Haginyan”, changes size in the PC display depending on the user’s beta/theta wave ratios. These ratios were chosen based on their potential as biomarkers of development (healthy children [15], young adults [16]), developmental disorders (e.g., autism spectrum disorders (ASD) [17], attention-deficit/hyperactivity disorders (ADHD) [18]), and intellectual disabilities [19]. To verify the applicability of the device and to visualize any unique NFT function, we looked for age-dependent changes in EEG signals in visitors at a local children’s play event, the “Wakuwaku Kid’s Festa”, where children were invited to experience the game.

The NFT game was made available in a small indoor space during the play event. An adult male instructor was present to help with the attachment of the EEG device and to explain its use. The game included three stages of visual discrimination tasks: Stage 1, a consecutive number task that induced neurocognitive activation; Stage 2, an open-eye resting condition; and Stage 3, the NFT task. Stages 1 and 2 were used as controls for comparison to Stage 3. To visualize relative changes in function-dependent NFT activity, Stage 1 was designed as the previously well-reported visual cognition but EEG-independent task. In contrast, Stage 2 was defined as another control task, “resting”, with active visual function (opening the eyes) but no cognitive function. Acquired EEG data were analyzed to determine any age-related differences using linear regression analysis [20] in 2 dimension plots of each EEG brain wave marker versus age.

## 2. Materials and Methods

### 2.1. Participant Age and Sex

Several hundred children and adults attended the “Wakuwaku Kids Festa” in Hagi city on 20 March 2021. Over a 3-h period, 36 people played the NFT game. The age and gender of each player were recorded. Of the 36 players, 6 were removed from the analysis; 2 because they were older than 30 years, and 4 because their EEG data were particularly noisy. The children who were removed were one 3-year-old, one 4-year-old, and two 7-year-olds. Table 1 shows the age and sex of the 30 remaining participants.

### 2.2. EEG Measurement System

#### 2.2.1. Hardware

A prototype EEG electrode headset was fabricated for this study (Figure 1). Two EEG scalp electrodes (approved medical devices, Japanese notification number 13B2 × 10278000001) were placed in a headband to be worn on the forehead. The headband allowed quick (10–15 s) and easy placement of Fp1 and Fp2 of the international 10–20 system [14] by the subject. Occasionally minor placement adjustments were required by the instructor. The electrodes were attached with disposable solid gel electrode patches (intercross-810: Intercross Corporation, Tokyo, Japan) to stabilize electrode-to-electrode resistance [21]. Electrophysiological reference and ground electrodes were clip-on silver chloride electrodes placed on the left and right ears, respectively. The headset was equipped with a commercially available 38 g small EEG module (intercross-413 Intercross Co., Japan) connecting all the electrodes (Figure 1).

To play, the participant sits looking toward a small projector displaying PC images sized 75 × 50 cm on the wall approximately 1 m away. Instructions for the game are displayed on the screen at the beginning of each stage.

#### 2.2.2. Signal Pre-Processing

EEG signals obtained from the GEL electrodes were input to the monitoring module (Figure 1) for 12-fold amplification, converted to digital data by 500 Hz, 24-bit AD conversion (integrated with time data by the electroencephalograph’s internal clock), and transferred to a Windows PC via Bluetooth. The transferred EEG signals were first processed with digital filtering High-Pass above 2 Hz with the Butterworth second-order filter to exclude body motion errors. The Low-Pass filter was digitally operated under 40 Hz by a Butterworth 8th order filter.

### 2.3. Three Stages of Visual Tasks, “0back”, “Rest”, and “NFT”

We designed a visual task with three stages (Figure 2). The first: 0BACK; the second: REST; the third: NFT sequentially, without any intervals. Each stage lasted 30 s with a 5 s interval between stages. The first two stages were reference tasks, and the third stage was the principal task, NFT. Using customized software (C#, Microsoft, Redmond, DC, USA), the tasks were synchronized to regulate the EEG monitoring software by trigger signals at the beginning and the end of each stage.

The first reference stage, “0back”, was designed as a neurocognitive activation task requiring sustained attention and concentration. Numerical values from 1 to 5 were randomly displayed and replaced every 0.6 s in the 15 × 15 cm area of the wall screen. The player began the task with their hands open and was instructed to close both hands as soon as the “5” was displayed. Given the simplicity of this task for all players, the accuracy evaluation was omitted.

In the second reference stage, “REST”, in a relaxed open-eye condition, the player passively viewed a non-arousing display image, Wall 2.jpg, from the Open Affective Standardized Image Set (OASIS; the minimum score for arousal index, 1.693069307) [22].

The last stage was the principal NFT task. In this task, the player was asked to interactively change the size of a familiar illustrated character (Haginyan, the Hagi City mascot) that was shown on the screen. In this prototype, the size of Haginyan increased or decreased based on the player’s beta/theta ratios relative to their calculated reference value (average of the beta/theta ratios from each player’s two reference stages) [23].

### 2.4. Analyses of Age-Dependency in Relative Power of Four EEG Wave Bands

The acquired EEG data were divided into frames per second. Frames with waveforms of 80 μV or higher were regarded as noise and were excluded. Each remaining frame underwent frequency analysis by Discrete Fourier Transform (DFT) [24]. The power values and content ratios per stage were calculated for each frequency band, delta, theta, alpha, and beta (Table 2) from the obtained DFT results. The trend graph of the four EEG bands and the beta/theta ratios were seen only in real-time by the instructor to confirm the success of the trial. To assure equality of data across subjects and to avoid missing data, a 10-s period (from 20–30 s of each Stage, 1 and 2) of the “0back” and “Rest” stages were analyzed, and a 25-s period (from 5–30 s) of the “NFT” stage was analyzed.

To compare between stages or SP1, left (L) and Sp2, right (R), the following equation with median values of each wave band power was used for the ratio variables to be plotted.

[Comparison in Stages (0BACK, REST, NFT)]

ARatio (0BACK versus REST) = (0BACK − REST)/(0BACK + REST)BRatio (NFT versus REST) = (NFT − REST)/(NFT + REST)CRatio (NFT versus 0BACK) = (NFT − 0BACK)/(NFT + 0BACK)

[Comparison in Left and Right brain sides (Sp1, Sp2)]

LR difference Ratio: LRD (Sp1 versus Sp2) = (Sp1 − Sp2)/(Sp1 + Sp2)

After visualizing each scatter plot with age as the independent variable on the horizontal axis and the average of each band index as the dependent variable on the vertical axis, a linear single regression analysis was performed to evaluate if the age-dependent model was significant using statistical software R, ggplot [25]. The significance was indicated as * (*p* < 0.05) and (*) (nearly *p* = 0.05).

## 3. Results

Of the 36 visitors (including 33 children) to play our NFT game, all could mount the headset easily within ten seconds and appeared to enjoy the game. Even though none of the participants had prior experience with the game, all completed all three stages without stopping. EEG signal acquisition was successful for all participants, but data from four children were omitted from the analysis because of noise in the post-analysis.

### 3.1. Trends of Four EEG Wave Band Ratios during Three Stages of the Game

Figure 3 shows two 10-s trend examples per stage of two subjects, an 8-year-old male and a 12-year-old female. The upper panel shows the power value from the DFT results for each frequency band, delta, theta, alpha, and beta, integrated from the bottom to the top, and the lower panel shows the change in the beta/theta for each left (Sp1) or right (Sp2) forehead electrode record. Results from Sp1 and Sp2 were generally synchronized within individuals but varied by individual, age, stage, and wave bands.

### 3.2. Age-Dependent Linear Regression Analysis

#### 3.2.1. Four Wave Band Power Content and Beta/Theta Ratios

In the age-dependent linear regression analysis of EEG wave bands and beta/theta ratios (Figure 4 and Table 3), no results were statistically significant, though an age-dependent decrease in delta at Sp1 approached significance (*p* = 0.052, enclosed by a red square line in Figure 4). In general, though not significant, the age-dependent linear regression model patterns in Sp1 looked weakly similar across the three stages, whereas Sp2 was more diverse.

The horizontal axis; age [y; year], the vertical axis; each wave ratio in four wave bands (delta, theta, alpha, beta) [%] and the beta/theta ratio. A pentagram or a parenthetic pentagram shows *p* < 0.05 or nearly *p* = 0.05 by linear regression analysis (blue lines), respectively.

#### 3.2.2. Median Value Comparison between Stages

When comparing median values between stages, delta band power was significantly higher in NFT compared to REST at Sp2 (Figure 5B: NFT versus REST, enclosed by a red box, and Table 4, *p* = 0.030 *) with other delta and theta lines showing weakly similar trends.

A pentagram or a parenthetic pentagram shows *p* < 0.05 or nearly *p* = 0.05 by linear regression analysis (blue lines), respectively.

There was a trend towards an age-dependent increase in Sp1 alpha in NFT compared to 0BACK (Figure 5C: NFT versus 0BACK, enclosed by a black box, and Table 4, *p* = 0.085 (*)). In addition, the age-dependent linear regression lines of 0BACK versus REST (Figure 5A) showed a general trend of OBACK exceeding REST, whereas no trend was seen in fast waves, beta, and alpha, of NFT versus REST (Figure 5B).

The horizontal axis; age [y; year], the vertical axis; each wave ratio between each pair of stages. An asterisk or a parenthetic asterisk shows *p* < 0.05 or nearly *p* = 0.05 by linear regression analysis, respectively.

#### 3.2.3. Median Value Comparison between Left (Sp1) and Right (Sp2)

Finally, we explored the lateralization of brain functional dominance. We found an age-dependent trend toward left-brain dominance in beta waves during the NFT stage (Figure 6 beta and NFT, the graph enclosed by a red square, Table 5, *p* = 0.052 (*)). The NFT trigger signals, beta/theta ratios at NFT also showed a similar weak trend toward left dominance (*p* = 0.087 (*), enclosed by a black square in Figure 6).

A pentagram or a parenthetic pentagram shows *p* < 0.05 or nearly *p* = 0.05 by linear regression analysis (blue lines), respectively.

Another fast wave band, alpha, exhibited significant age-dependent left lateralization at the REST stage (Figure 6 alpha REST, Table 5. *p* = 0.044).

The horizontal axis; age (y; year), the vertical axis; each wave ratio between Left (Sp1) and Right (Sp2) electrodes. An asterisk or a parenthetic asterisk shows *p* < 0.05 or nearly *p* = 0.05 by linear regression analysis, respectively.

## 4. Discussion

A home-based, easy-to-use EEG system to monitor brain development and assist with clinical diagnosis and treatment in children has long been promised [26]. Various EEG devices and systems have been recently released [27], though the widespread utility of these devices suffers due to the complicated requirements associated with mounting and wearing a delicate EEG device on the head and difficulty in performing meaningful analysis of the obtained data. This study addressed these challenges by developing an easy-to-deploy headband-mounted EEG device designed for children with gel electrode patches, Bluetooth EEG, and an NFT-based interactive PC game and associated analysis software. We evaluated whether children would readily use this system at a one-day local children’s event. We further explored whether there were any age-dependent EEG signal differences that would reproduce findings of previous reports [7,17,19,28].

Thirty-six individuals completed the experiment. Of these, a 3-year-old male and a 4-year-old female’s EEG data were too noisy to be analyzed. If this availability might be for 5 years and older, 32 were 5 to 15 years old. Of the 32, EEG data from 30 subjects were deemed appropriately consistent for analysis, indicating the success of this preliminary prototype design. The experiment consisted of three stages. The first two stages, 0BACK and REST (which were known to change in an age-dependent manner [2,4]), were used for reference and did not involve any feedback from the subjects. These stages were designed with an aspect of comparative baselines. In the final stage, NFT, the size of the projected image changed in response to the subject’s own beta/theta ratios relative to their value during the reference stages. This was the first experience with such technology for all subjects. If the function could be arbitrarily controlled, it might lead to the development of novel neurological function, much like a magic ability to change images with “telekinesis”. This may be one potential core technology that could be applied to future work on brain–computer interfaces [29]. Because the NFT stage was limited to 30 s in this study, we did not evaluate learning and neuronal plasticity [30,31], but rather, we focused on acute and real-time responses. For this reason, we did not evaluate the control status of the NFT but plan to do so in future studies.

In our age-dependent linear regression models, we found some significance in delta, alpha, and beta waves and left-right or stage differences. In general, there was more age dependence in NFT and REST stages, close-eyes [2] and open-eyes [32], and left lateralization than in the visual cognition task, 0BACK [33]. We found an age-dependent decrease in Sp1 delta at stage REST along with other band trends that were consistent with previous reports, indicating that our device functioned correctly and that the children were similar to the previously reported population. We also found age-dependent left-dominance lateralization [34], age-dependent NFT dominance versus REST at Sp2 delta, and a weak age-dependent 0BACK dominance versus NFT at Sp1 alpha. Our study population had a strong female bias; thus, the study population should be enhanced in both age and sex to allow determination regarding the generalizability of our models.

Although this study for child use selected a simplified strategy of forehead Sp1 and Sp2 channels in the 10–20 method, the collectible neuronal information was restricted. Multiple electrodes or other placement extensions should be designed to reach more complex functions such as neuropsychological or cognitive development with another supporting headset such as a cap or headphone style [35].

This child-friendly system holds promise for helping to address the recent increase in neurodevelopmental disorders. Regarding diagnostics, EEG-determined brain waves and theta/beta ratio have potential benefits as a biomarker for ADHD and ASD [36]. Children with ADHD have decreased beta waves and increased theta waves, which is believed to be correlated with poor attention [37]. Furthermore, this fun-to-use system has the potential to assist in development through its longitudinal use in educational settings [38,39].

As with other aspects of therapeutic and rehabilitative education for developmental disorders, the combination of EEG and digital tools, such as virtual and augmented reality devices and game software, have been recently suggested to lead the evolution of treatment [40].

Clinical use of NFT in children was reported in ADHD [41], ASD [17], anxiety disorders, and depression [28]. In addition to neurofeedback applications, EEG itself may enable predictive diagnosis of neurodevelopmental disorders in early ages [42], given that resting EEG correlates with neurochemical imaging MRS [43], the default mode network [44], visual [45,46], auditory [47], tactile [48], and subcortical networks [42] in autism [49].

Since the enactment of the Developmental Disabilities Support Law in Japan in 2005, there has been increased attention paid to early diagnosis of developmental disabilities, developmental support within schools, and building support centers [50]. In 2012, the “Survey on Students with Special Educational Needs with Developmental Disabilities in Regular School Classes” conducted by the Ministry of Education, Culture, Sports, Science, and Technology, reported that 6.5% of students are estimated to have developmental disabilities, and the number has increased significantly over the past decade (https://www.soumu.go.jp/menu_news/s-news/110614.html, accessed on 31 July 2022). Finding simple and scalable neurodevelopmental biomarkers for early detection of developmental disabilities is needed [51] because neurodevelopmental symptoms may precede behavioral symptoms by months to years, allowing for early intervention to perhaps prevent negative outcomes [52], such as withdrawal from school. Thus, our approach to develop an EEG system for children involving interactive games that can be implemented in a home or child-friendly social environment indoors and outdoors would be an important contribution to current education practices [53]. We believe that the easy-to-use design of our prototype is more important in NFT than the additional information that could be obtained using more restrictive techniques. Extension of novel designs of NFT would be expected by this simplified attachable EEG.

## 5. Conclusions

The Sp1 and Sp2 channel EEG headsets we designed to support children’s general and clinical education took less than 10 s to be properly placed by the individuals and were successfully worn by 32 children aged 5 to 15 years. All subjects appeared to enjoy the three stages of the task, identifying the numbers on the display, open-eyes resting, and using neurofeedback to adjust the size of the local mascot image according to the beta/theta ratio. Post analysis of the recorded EEG data suggested a moderate age-dependent left dominance lateralization in the beta wave band or relatively age-dependent dominance versus open-eyes resting at Sp2. These models need further testing.

## Figures and Tables

**Figure 1 sensors-22-07207-f001:**
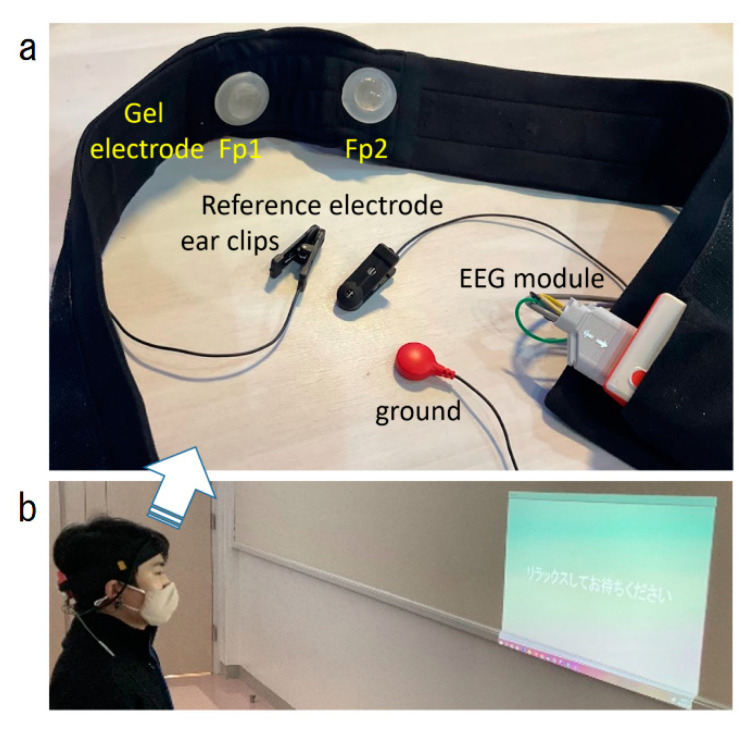
Headset of EEG device (**a**), a player wearing the headset and looking at the game display (**b**).

**Figure 2 sensors-22-07207-f002:**
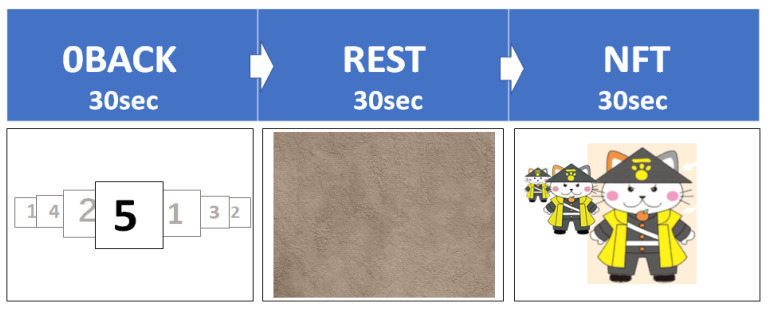
The three stages of the game protocol.

**Figure 3 sensors-22-07207-f003:**
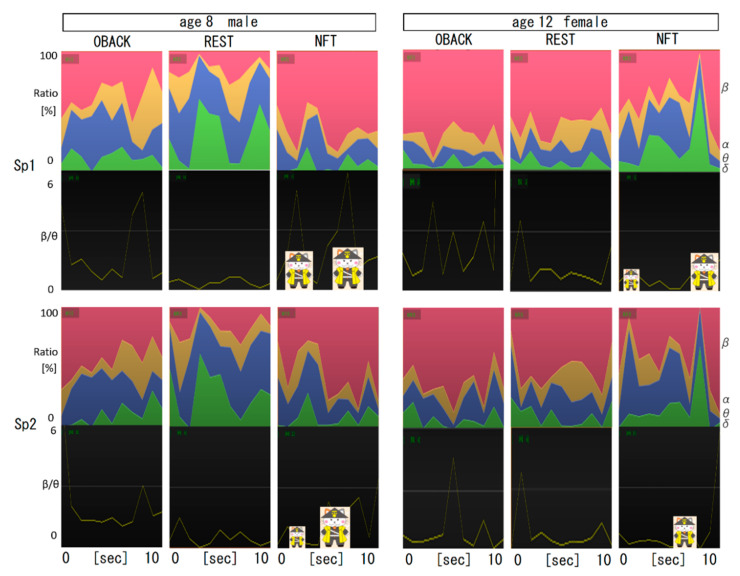
Examples of the EEG trend of four band and beta/theta ratios per stage, 0BACK, REST, and NFT in younger and older subjects left (Sp1) or right (Sp2) brain.

**Figure 4 sensors-22-07207-f004:**
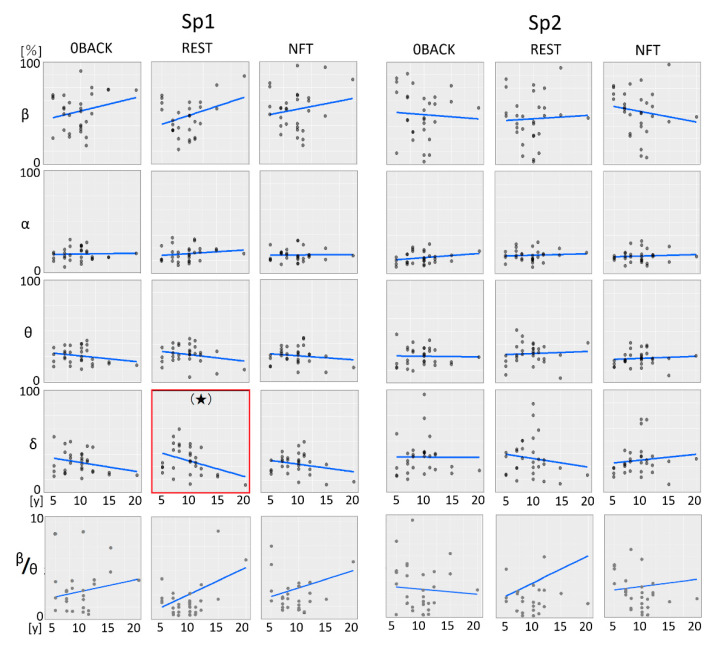
Age-dependent comparison of wave bands and beta/theta in stages and channels.

**Figure 5 sensors-22-07207-f005:**
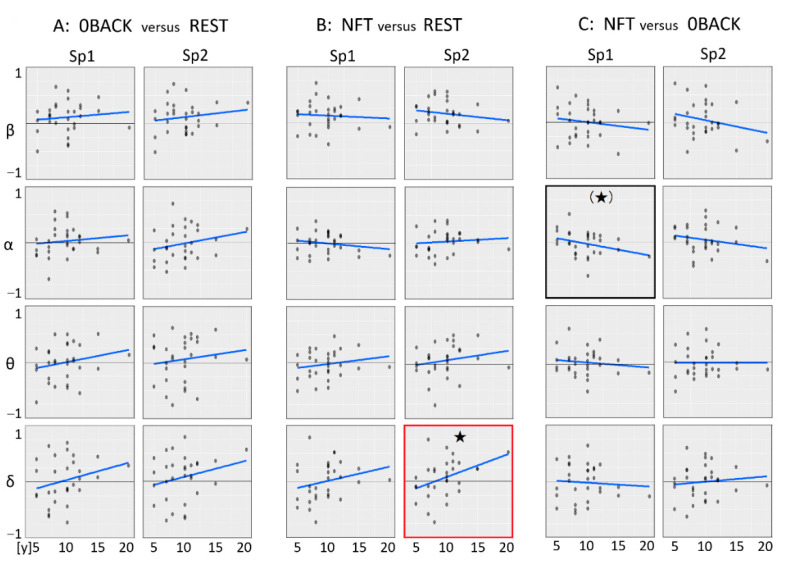
Each median of channel and wave band comparisons between stages.

**Figure 6 sensors-22-07207-f006:**
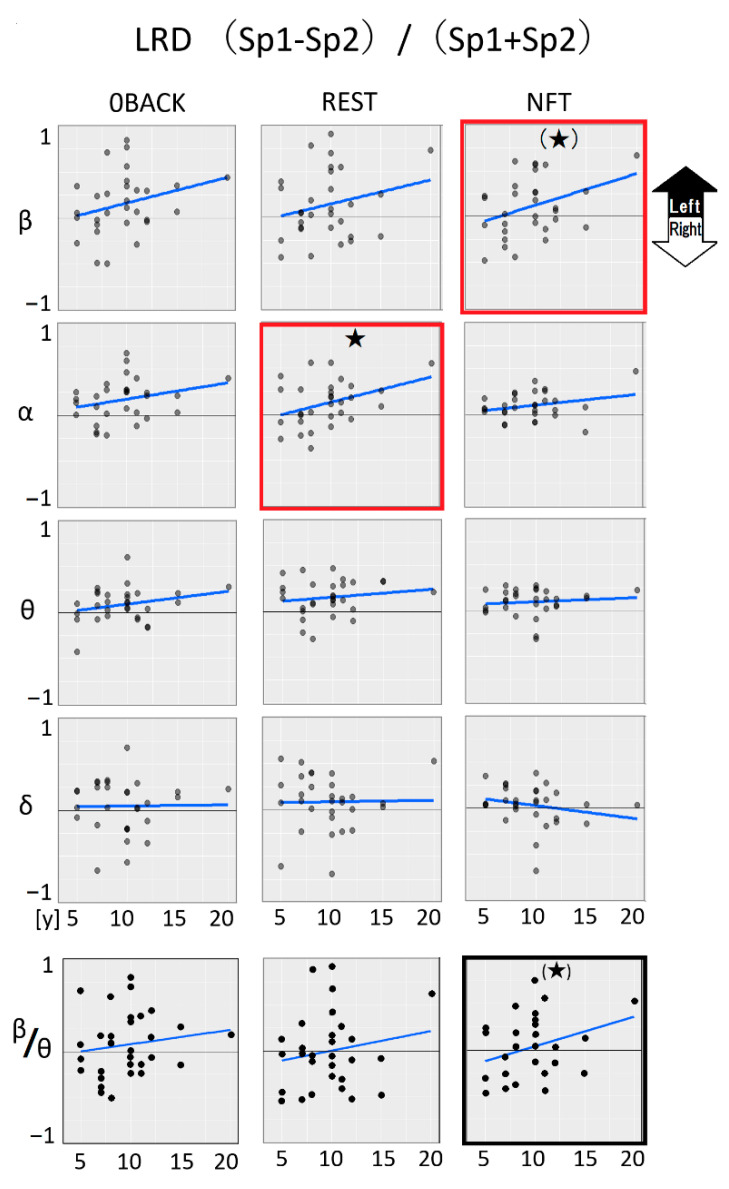
Left or right brain dominance: age-dependent lateralization.

**Table 1 sensors-22-07207-t001:** Number and ages of participants.

Age [year]	*n* (Female)	*n* (Male)	*n* (Total)
5	1	3	4
7	5	-	5
8	-	4	4
10	7	1	8
11	3	-	3
12	3	-	3
15	2	-	2
20	1	-	1
Total [*n*]	22	8	30
Average [year]	10.4	7.1	9.5
SD [year]	3.2	1.8	3.3

**Table 2 sensors-22-07207-t002:** The wave bands of EEG.

Band	Range [Hz]
delta	2~4
theta	4~8
alpha	8~13
beta	13~30

**Table 3 sensors-22-07207-t003:** *p*-values associated with age-dependent EEG wave linear regression shown in Figure 4.

		Sp1			Sp2	
	0BACK	REST	NFT	0BACK	REST	NFT
beta	0.209	0.121	0.359	0.758	0.805	0.407
alpha	0.842	0.405	0.921	0.216	0.662	0.675
theta	0.230	0.228	0.423	0.911	0.732	0.643
delta	0.169	**0.052 (*)**	0.187	0.980	0.439	0.515
beta/theta	0.351	0.132	0.301	0.722	0.386	0.710

(*), bold: nearly *p* = 0.05.

**Table 4 sensors-22-07207-t004:** *p*-values associated with age-dependent EEG wave linear regression shown in Figure 5.

	A: 0BACK Versus REST		B: NFT Versus REST		C: NFT Versus 0BACK	
	Sp1	Sp2	Sp1	Sp2	Sp1	Sp2
beta	0.557	0.406	0.718	0.333	0.415	0.194
alpha	0.508	0.221	0.33	0.591	**0.095 (*)**	0.279
theta	0.237	0.459	0.318	0.299	0.495	0.993
delta	0.17	0.21	0.196	**0.030 ***	0.709	0.575

*, bold: *p* < 0.05, (*), bold: nearly *p* = 0.05 (*p* < 0.1).

**Table 5 sensors-22-07207-t005:** *p*-values associated with age-dependent EEG wave linear regression shown in Figure 6.

	LRD(Sp1 − Sp2)/(Sp1 + Sp2)		
	0BACK	REST	NFT
beta	0.136	0.214	**0.052 (*)**
alpha	0.158	**0.044 ***	0.157
theta	0.183	0.423	0.59
delta	0.941	0.924	0.246
beta/theta	0.433	0.356	**0.087 (*)**

*, bold: *p* < 0.05, (*), bold: nearly *p* = 0.05 (*p* < 0.1).

## Data Availability

The datasets generated for this study are available on request to the corresponding authors.

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
