# Peer review of "Simplified Attachable EEG Revealed Child Development Dependent Neurofeedback Brain Acute Activities in Comparison with Visual Numerical Discrimination Task and Resting"

_sensors, 2022, doi:10.3390/s22197207_

Round 1
Reviewer 1 Report
In this study, a two-channel EEG headband measurement device has been utilized to perform neurofeedback analysis relevant to education or neurocognitive development. The methods and results reported in the paper might contribute to the use of EEG signals to perform neurofeedback analysis. Below are some comments that aim to improve the paper.
1. Please summarize the numerical results obtained in your study at the end of the abstract.
2. Page 1, lines 26-27: the two lines are not connect with the abstract. I am not sure what is the purpose of these two lines.
3. Introduction section: Please expand the literature review by adding more recent studies that are related to the use of EEG analysis to perform neurofeedback analysis in children.
4. Discussion section: can you elaborate on the potential of employing the outcomes of your study to improve the use of EEG-based devices to achieve effective neurofeedback analysis in children.
5. Discussion section: Can you compare the results reported in your study with the results reported in similar previous studies (which might be obtained using other EEG acquisition devices). How can the results reported in you study be considered as a contribution compared to the results reported in the literature.
Author Response
Dear Reviewer 1,
First, thank you for your meaningful comments and suggestions.
We revised this with a native US English speaking co-author.
In this study, a two-channel EEG headband measurement device has been utilized to perform neurofeedback analysis relevant to education or neurocognitive development. The methods and results reported in the paper might contribute to the use of EEG signals to perform neurofeedback analysis. Below are some comments that aim to improve the paper.
- Please summarize the numerical results obtained in your study at the end of the abstract.
We added the following sentences at the end of abstract.
Preliminary linear regression analyses by least-squares method of the acquired EEG and age data in thirty participants from 5 to 20 year-old suggested an age-dependent left brain lateralization of beta waves at the neurofeedback stage (p=0.052) and of alpha waves at the open-eye resting stage (p=0.044) with potential involvement of other wave bands. These results require further validation.
- Page 1, lines 26-27: the two lines are not connect with the abstract. I am not sure what is the purpose of these two lines.
We excluded the two lines from the manuscript.
- Introduction section: Please expand the literature review by adding more recent studies that are related to the use of EEG analysis to perform neurofeedback analysis in children.
We added to the second paragraph of the Introduction as follows;
NFT has been applied less in typically developing children but rather [6] it has been used clinically in children to both diagnose and provide therapy [7] for ADHD [8] using NFT theta, alpha, sensorimotor (SMR), and beta wave training [9], and for children in the autistic spectrum to improve cognitive function impairment [10] and in learning disorders [11] .
[6] F. W. & D. C. Wenya Nan, Mengqi Wan, Yali Jiang, Xiaoping Shi, “Alpha/Theta Ratio Neurofeedback Training for Attention Enhancement in Normal Developing Children: A Brief Report,” 47, pages 223–229.
[7] E. A. Shereena, R. K. Gupta, C. N. Bennett, K. J. V. Sagar, and J. Rajeswaran, “EEG Neurofeedback Training in Children With Attention Deficit/Hyperactivity Disorder: A Cognitive and Behavioral Outcome Study,” Clin. EEG Neurosci., vol. 50, no. 4, pp. 242–255, 2019.
[8] A. Lenartowicz and S. K. Loo, “Use of EEG to Diagnose ADHD,” Curr. Psychiatry Rep., vol. 16, no. 11, pp. 1–19, 2014.
[9] T. S. Moriyama, G. Polanczyk, A. Caye, T. Banaschewski, D. Brandeis, and L. A. Rohde, “Evidence-Based Information on the Clinical Use of Neurofeedback for ADHD,” Neurotherapeutics, vol. 9, no. 3, pp. 588–598, 2012.
[10] L. Mekkawy, “Efficacy of neurofeedback as a treatment modality for children in the autistic spectrum,” Bull. Natl. Res. Cent., vol. 45, no. 1, 2021.
[11] B. J. Martínez-Briones, J. Bosch-Bayard, R. J. Biscay-Lirio, J. Silva-Pereyra, L. Albarrán-Cárdenas, and T. Fernández, “Effects of neurofeedback on the working memory of children with learning disorders-an EEG power-spectrum analysis,” Brain Sci., vol. 11, no. 7, 2021.
- Discussion section: can you elaborate on the potential of employing the outcomes of your study to improve the use of EEG-based devices to achieve effective neurofeedback analysis in children.
We added the last second paragraph as follows;
Regarding diagnostics, EEG determined brain waves and theta/beta ratio have potential benefit as a biomarker for ADHD and ASD [35]. Children with ADHD have decreased beta waves and increased theta waves, which is believed to be correlated with poor attention [36]. Furthermore, this fun to use system has the potential to assist in development through its longitudinal use in educational settings [37] [38].
- Discussion section: Can you compare the results reported in your study with the results reported in similar previous studies (which might be obtained using other EEG acquisition devices). How can the results reported in you study be considered as a contribution compared to the results reported in the literature.
As mentioned above, our child friendly system can evaluate both neuronal activation and developmental progress simultaneously. We believe that our revised version successfully conveys the importance of this work with respect to the existing literature.

Reviewer 2 Report
A paper with potential, however comments affecting the sequence would better be targetted:
- mentioning the word sphenoidal electrodes in the introduction before going to sp1, sp2.
-section 2.2 needs more details: separate the hardware details from the pre-processing performed to transfer the signal. Detail why each of these decisions occurred, I couldn't gather why this exact setup was used?
- what was the goal for the rest stage ?
- requiring more details about the NFT stage itself, explaining what was done/ game play would help future researchers set reference
-I would reconsider adding some of the recent related work on the benefits of using EEG signals and divide them into sections based on usage: diagnosis or therapy/rehabitation. a suggestion is presented in the following reference that I cam across while searching:
Barba, Maria Cristina, et al. "BRAVO: a gaming environment for the treatment of ADHD." International conference on augmented reality, virtual reality and computer graphics. Springer, Cham, 2019.
-section 2.3 : why is the comparison conducted between the stages? It wasn't clear for me, I think a diagram explaining the exact process from the beginning would be very beneficial in this case.
-the first time we hear about a regression analysis being conducted, and is being a central part of the results is just before the results section. please explain the objectives in the paper clearly so as readers we can know what to expect from the results
-axes names for plots??
- the discussion section is one of the clear sections within the paper
Author Response
Dear Reviewer 2,
First, thank you for your meaningful comments and suggestions.
We revised this with a native US English speaking co-author.
A paper with potential, however comments affecting the sequence would better be targetted:
- mentioning the word sphenoidal electrodes in the introduction before going to sp1, sp2.
We added the following sentences and reference before the suggested part [10].
Page 2 from line 53
Recordings from sphenoidal electrodes have been suggested to improve the localization precision of EEG recordings in patients with temporal lobe epilepsy. However, the usage was limited [13].
Reference [10] M. B. Hamaneh, C. Limotai, and H. O. Lüders, “Sphenoidal electrodes significantly change the results of source localization of interictal spikes for a large percentage of patients with temporal lobe epilepsy,” J. Clin. Neurophysiol., vol. 28, no. 4, pp. 373–379, 2011.
-section 2.2 needs more details: separate the hardware details from the pre-processing performed to transfer the signal. Detail why each of these decisions occurred, I couldn't gather why this exact setup was used?
We separated this information into two subsections; 2.2.1. Hardware and 2.2.2. Signal pre-processing. We also provided a more detailed description of the signal pre-processing as follows;
Page 4 from line 1
2.2.2. Signal pre-processing
EEG signals obtained from the GEL electrodes were input to the monitoring module (Figure 1) for 12-fold amplification, converted to digital data by 500 Hz, 24-bit AD conversion, (integrated with time data by the electroencephalograph's internal clock,) and transferred to a windows PC via Bluetooth. The transferred EEG signals were first processed with digital filtering High-Pass above 2 Hz with the Butterworth second order filter to exclude body motion errors. The Low-Pass filter was digitally operated under 40 Hz by a Butterworth 8th order filter.
- what was the goal for the rest stage ?
In the second paragraph of the Discussion, we explained that the stage 1 ”0BACK” and stage 2 ”REST” were references to confirm similarity to the previous report about age-dependent changes [4] [2] and did not involve any feedback to the subjects in this study. In the introduction we now include an additional explanation of these stages as follows;
Page 2, the last paragraph of Introduction.
Stages 1 and 2 were used as controls for comparison to Stage 3. To visualize relative changes in function-dependent NFT activity, Stage 1 was designed as the previously well-reported visual cognition but EEG-independent task. In contrast, Stage 2 was defined as another control task “resting” with active visual function (opening the eyes) but no cognitive function.
- requiring more details about the NFT stage itself, explaining what was done/ game play would help future researchers set reference
We added the meaning of NFT function to humans as follows in the second paragraph of Discussion;
This was the first experience with such technology for all subjects. If the function could be arbitrarily controlled, it might lead to development of novel neurological function, much like a magic ability to change images with “telekinesis”. This may be one of potential core technology that could be applied to future work on brain computer interfaces [24].
[24] J. Sun, J. He, and X. Gao, “Neurofeedback Training of the Control Network Improves Children’s Performance with an SSVEP-based BCI,” Neuroscience, vol. 478, pp. 24–38, 2021.
-I would reconsider adding some of the recent related work on the benefits of using EEG signals and divide them into sections based on usage: diagnosis or therapy/rehabitation. a suggestion is presented in the following reference that I cam across while searching:
Barba, Maria Cristina, et al. "BRAVO: a gaming environment for the treatment of ADHD." International conference on augmented reality, virtual reality and computer graphics. Springer, Cham, 2019.
In the last 2nd paragraph, we added two divided explanation parts about diagnose and education therapy aspects before ones of NFT as follows;
This child friendly system holds promise for helping to address the recent increase in neurodevelopmental disorders. Regarding diagnostics, EEG determined brain waves and theta/beta ratio have potential benefit as a biomarker for ADHD and ASD [35]. Children with ADHD have decreased beta waves and increased theta waves, which is believed to be correlated with poor attention [36]. Furthermore, this fun to use system has the potential to assist in development through its longitudinal use in educational settings [37] [38].
As with other aspects of therapeutic and rehabilitative education for developmental disorders, combination of EEG and digital tools such as virtual and augmented reality devices and game software, have been recently suggested to lead evolution of treatment [32].
Clinical use of NFT in children were reported in ADHD [33], ASD [14], Anxiety disorders and depression [23]. In addition to neurofeedback applications, EEG itself may enable predictive diagnosis of neurodevelopmental disorders in early ages [34] given that resting EEG correlates with neurochemical imaging MRS [35], the default mode network [36], visual [37][38], auditory [39], tactile [40], and subcortical networks [34] in autism [41].
-section 2.3 : why is the comparison conducted between the stages? It wasn't clear for me, I think a diagram explaining the exact process from the beginning would be very beneficial in this case.
The reason to compare the three stages was mentioned and additional explanation has been added to the text. In section 2.3, we have added a short explanation into the current sentences as follows;
We designed a visual task with three stages, 1st: 0BACK, 2nd: REST and 3rd: NFT, sequentially, without any intervals.
-the first time we hear about a regression analysis being conducted, and is being a central part of the results is just before the results section. please explain the objectives in the paper clearly so as readers we can know what to expect from the results
We added the following information to the last part of Introduction:
Acquired EEG data were analyzed to determine any age-related differences using a linear regression analysis [20] in 2 dimension plots of each EEG brain wave marker versus age.
-axes names for plots??
Axis labels are now included in each figure legend as follows;
Figure 4. Age-dependent comparison of wave bands and beta/theta in stages and channels
The horizontal axis; age [y; year], the vertical axis; each wave ratio in four wave bands (delta, theta, alpha, beta) [%] and the beta/theta ratio. An asterisk or a parenthetic asterisk shows p<0.05 or nearly p=0.05 by linear regression analysis, respectively.
Figure 5. Each median of channel and wave band comparisons between stages.
The horizontal axis; age [y; year], the vertical axis; each wave ratio between each pair of stages. An asterisk or a parenthetic asterisk shows p<0.05 or nearly p=0.05 by linear regression analysis, respectively.
Figure 6. Left or right brain dominance: age-dependent lateralization
The horizontal axis; age [y; year], the vertical axis; each wave ratio between Left (Sp1) and Right (Sp2) electrodes. An asterisk or a parenthetic asterisk shows p<0.05 or nearly p=0.05 by linear regression analysis, respectively.
- the discussion section is one of the clear sections within the paper
We appreciate the reviewer’s positive comment.

Round 2
Reviewer 1 Report
I recommend to accept the paper in the current format.
Reviewer 2 Report
the comments were mostly answered sufficinetly , however please revise the writing for the added part for section 2.3. I figure you mean the stages were added as a comparative baseline?
Author Response
Thank you for your review comment.
Comments and Suggestions for Authors
the comments were mostly answered sufficinetly , however please revise the writing for the added part for section 2.3. I figure you mean the stages were added as a comparative baseline?
We think the first two stages as reference tasks because we focused on not only NFT but also these two stages. The study design was not limited for comparative baseline. Instead, in accordance with the reviewer's comments, we added the green text on Lines 321-322 of Discussion,.
"The experiment consistent of three stages. The first two stages, 0BACK and REST (which were known to change in an age-dependent manner [4] [2]), were used for reference and did not involve any feedback to the subjects. These stages were designed with an aspect of comparative baseline."
In addition, we found an error of subsection number to be corrected from 2.3 (error) to 2.4 (correct) on Line 152.
